# Functional Expression of Piezo1 in Dorsal Root Ganglion (DRG) Neurons

**DOI:** 10.3390/ijms21113834

**Published:** 2020-05-28

**Authors:** Jueun Roh, Sung-Min Hwang, Sun-Ho Lee, Kihwan Lee, Yong Ho Kim, Chul-Kyu Park

**Affiliations:** 1Gachon Pain Center and Department of Physiology, College of Medicine, Gachon University, Incheon 21999, Korea; jueun9392@gmail.com (J.R.); unclehwang76@gmail.com (S.-M.H.); key1479@gmail.com (K.L.); 2Department of Neurosurgery, Spine tumor center, Samsung Medical Center, Sungkyunkwan University School of Medicine, 81 Irwon-ro, Gangnam-gu, Seoul 06351, Korea; sobotta72@gmail.com

**Keywords:** Piezo, dorsal root ganglion, pain, TRPV1, mechanotransduction

## Abstract

Piezo channels are mechanosensitive ion channels. Piezo1 is primarily expressed in nonsensory tissues, whereas Piezo2 is predominantly found in sensory tissues, including dorsal root ganglion (DRG) neurons. However, a recent study demonstrated the intracellular calcium response to Yoda1, a selective Piezo1 agonist, in trigeminal ganglion (TG) neurons. Herein, we investigate the expression of *Piezo1* mRNA and protein in mouse and human DRG neurons and the activation of Piezo1 via calcium influx by Yoda1. Yoda1 induces inward currents mainly in small- (<25 μm) and medium-sized (25–35 μm) mouse DRG neurons. The Yoda1-induced Ca^2+^ response is inhibited by cationic channel blocker, ruthenium red and cationic mechanosensitive channel blocker, GsMTx4. To confirm the specific inhibition of Piezo1, we performed an adeno-associated virus serotype 2/5 (AAV2/5)-mediated delivery of short hairpin RNA (shRNA) into mouse DRG neurons. AAV2/5 transfection downregulates *piezo1* mRNA expression and reduces Ca^2+^ response by Yoda1. Piezo1 also shows physiological functions with transient receptor potential vanilloid 1 (TRPV1) in the same DRG neurons and is regulated by the activation of TRPV1 in mouse DRG sensory neurons. Overall, we found that Piezo1 has physiological functions in DRG neurons and that TRPV1 activation inhibits an inward current induced by Yoda1.

## 1. Introduction

Chemical and thermal-induced nociceptor responses in sensory neurons have been well studied [1]. In contrast, mechanotransduction in sensory neurons is not well understood. Mechanosensitive ion channels (MSCs) represent a critical class of mechanotransduction signals that directly and rapidly convert mechanical force into electrochemical signals [2]. The discriminative molecular identities of mechanotransduction for sensory signals such as touch perception, proprioception, and mechanical pain remained elusive until the discovery of the Piezo channel family, with its members Piezo1 and Piezo2 which mediate mechanically activated cationic currents in mammalian cells [3]. Piezo channels play a critical role in a variety of mechanotransduction processes [4,5]. In particular, Piezo1 is involved in embryonic vascular maturation, blood pressure regulation, urinary osmoregulation, epithelial homeostasis, neural stem cell fate determination, and axonal guidance, whereas Piezo2 mediates the sensation of gentle touch, proprioception, tactile allodynia, airway stretch, and lung inflation [4,6,7,8].

In the peripheral sensory system, Piezo channels are promising candidates for studying mechanical pain transduction [9]. Piezo2 is found in more than 45% of dorsal root ganglion (DRG) neurons [10,11,12] and has been extensively studied for its mechanotransduction function in primary sensory neurons [10,12,13]. However, some studies have shown that Piezo2 is either not or only partially involved in mechanical pain responses [10,11,14,15,16]. Another study has confirmed that human patients without *Piezo2* exhibit touch discrimination and joint proprioception and have a normal mechanical pain threshold [10,11,17,18]. DRG and trigeminal ganglia (TG) contain the cell bodies of highly specialized primary afferent neurons that transmit sensory information from the periphery to the central nervous system (CNS) [19]. A recent study has found that Piezo1 is expressed in the TRPV1-positive peptidergic nociceptive nerve fibers of the trigeminal ganglion in rats and mice, indicating novel migraine-related mechanotransduction pathways [13]. 

It is important to address the expression and function of Piezo1 activity in DRG neurons, due to the expression and physiological functions of Piezo1 in TG, which has similar properties to DRG. Currently, the methods used to record Piezo channel activity are fairly limited, as they do not allow easy discrimination between the activity of Piezo1 and Piezo2. Yoda1 is a Piezo1 agonist that can be used to easily measure the function of Piezo1 in various tissues, including sensory neurons. Recently, the Hamill group has reported that *Piezo1* mRNA is expressed in mouse DRG neurons that are distinct from neurons expressing TRPV1 [14]. Together, these findings suggest that further studies are needed to investigate the physiological functions of Piezo1 using Yoda1 in DRG neurons to better understand its role. 

In the present study, we have confirmed that Piezo1 is expressed in mouse and human DRG neurons. Functionally, Yoda1 induces an increase in intracellular calcium and an inward current through the activation of Piezo1 in small- and medium-sized DRG neurons. These physiological functions of Piezo1 are regulated by TRPV1 activation in DRG neurons in mice. Therefore, we have demonstrated an important new physiological role of Piezo1, in addition to the mechanosensitivity function of Piezo2 in DRG neurons.

## 2. Results

### 2.1. Physiological Function of Piezo1 in Mouse Dorsal Root Ganglion (DRG) Neurons

To identify the physiological functions of Piezo1 in mouse dorsal root ganglion (DRG) neurons, we used whole-cell patch-clamp recordings. Sequential applications of Yoda1 induced an inward current with a desensitization pattern in mouse DRG neurons (Figure 1A,B). The percentage of total DRG neurons that were responsive to Yoda1 was 29% (22/76), of which 24% (10/41) were small (<25 μm)-sized neurons, 50% (10/20) were medium (25–35 μm)-sized neurons, and 13% (2/15) were large (>35 μm)-sized neurons. Furthermore, we tested whether Yoda1 led to a rapid calcium increase through Piezo1 channels in mouse DRG neurons. Sequential applications of Yoda1 induced Ca^2+^ response in the presence of 2 mM extracellular Ca^2+^ (Figure 1C,D). Calcium imaging revealed that neurons responsive to Yoda1 were small-sized neurons (92.6%), while the remaining four neurons were medium-sized (7.4%). Extracellular Ca^2+^ removal abolished Yoda1-induced intracellular Ca^2+^ increase, in a reversible manner (Figure 1E,F). Therefore, these results suggest that Piezo1 can be pharmacologically activated in mouse DRG neurons and drives a cation influx. 

### 2.2. Piezo1 Expression in Mouse and Human Dorsal Root Ganglion (DRG) Neurons

Subsequently, we examined the expression of Piezo1 in mouse dorsal root ganglion (DRG) neurons. Reverse transcription-polymerase chain reaction (RT-PCR) and Western blots were used to assess *Piezo1* mRNA and protein expressions (Figure 2A,B). Additionally, *piezo2* and *trpv1* mRNA were expressed in DRG (Figure 2A). To further confirm the expression pattern of Piezo1 according to the size of DRG neurons, single-cell RT-PCR experiments were performed by collecting samples of DRG neurons that were allocated to three different groups based on their size (small, medium, and large). *piezo1* mRNA was detected in 18.5% of all DRG neurons (12/65), of which 13% (4/30), 33% (5/15), and 15% (3/20) were small-, medium-, and large-sized neurons, respectively (Figure 2C). These results confirmed that Piezo1 is widely expressed in mouse DRG neurons. *piezo2* mRNA was detected in 39% of all DRG neurons (22/56), of which 26% (7/27), 33% (3/9) and 60% (12/20) were small-, medium-, and large-sized DRG neurons. Next, we examined the physiological function of Piezo1 in human DRG neurons. RT-PCR analysis revealed that the *piezo1* gene was expressed in the DRG of two different humans (Figure 2D). Application of Yoda1 induced Ca^2+^ response in the presence of 2 mM extracellular Ca^2+^ in human DRG neurons (Figure 2E). 

### 2.3. Pharmacological Inhibition of Piezo1 in Mouse Dorsal Root Ganglion (DRG) Neurons

After confirming the normal physiological function of Piezo1 in mouse dorsal root ganglion (DRG), we assessed the functional changes in Piezo1 induced by various cation channel inhibitors. We treated DRG neurons with three inhibitors for 1 min before the second application of Yoda1. Ruthenium red (RR), a nonselective cation channel blocker, significantly inhibited Yoda1-induced Ca^2+^ increase (control = 0.91 ± 0.24; ruthenium red = 0.18 ± 0.03) (Figure 3A,B,E). Moreover, we used spider venom peptide that inhibits cationic mechanosensitive channels, GsMTx4, which almost completely blocked Yoda1-induced Ca^2+^ increase (control = 0.61 ± 0.11; GsMTx4 = 0.07 ± 0.01) (Figure 3A,C,E). Dooku1, which blocks Yoda1-induced Piezo1 activity, also dramatically inhibited Yoda1-induced Ca^2+^ increase (control = 0.74 ± 0.19; Dooku1 = 0.13 ± 0.03) (Figure 3A,D,E). Subsequently, we developed an effective method to knockdown Piezo1 expression in dissociated DRG neurons using AAV2/5-mediated short-hairpin RNA to more specifically inhibit *piezo1* gene expression. A Taqman assay confirmed that the knockdown of Piezo1 in mouse DRG resulted in significantly decreased expression (63%) of *piezo1* mRNA, compared to that of the control mouse DRG neuron (Figure 3F). Calcium imaging showed that the Yoda1-induced Ca^2+^ response was significantly inhibited by approximately 59% in the mouse DRG neurons with Piezo1 knockdown but not scramble (Figure 3G,H). 

### 2.4. Physiological Function of Piezo1 and TRPV1 in the Same Dorsal Root Ganglion (DRG) Neurons

Finally, we investigated whether Piezo1 is expressed in dorsal root ganglion (DRG) simultaneously with TRPV1. The sequential application of both Yoda1 and capsaicin induced an inward current in the same DRG neurons (Figure 4A). Ninety percent of neurons (19/21) expressing Piezo1 were found to express TRPV1 (Figure 4B). Furthermore, we tested the functional effect of Piezo1 by TRPV1 activation. Although the second application of Yoda1 induced currents that were desensitized compared to the first application of Yoda1 (Figure 4C), capsaicin completely inhibited the inward current of Piezo1 caused by the second application of Yoda1 (Figure 4B,E,F). 

## 3. Discussion

Mechanosensitive ion channels are expressed in various types of sensory neurons [20,21,22,23]. It has been reported that the detection of mechanical touch or forces are mediated by different ion channels such as potassium K2P channel, mechanosensitive ion channels (MSCs), TMEM 120A/TACAN, TMEM63/OSCA, TMC1/2, and Piezo1/2 in *D**rosophila*, plants, mice, and humans, respectively [24,25,26,27]. However, the identities of mechanotransduction channels in mammalian are largely unknown. It has recently been found that TACAN shares no sequence similarity with known Piezos and transmembrane channels like (TMC) protein [26]. TACAN is expressed predominately in small-diameter neurons of dorsal root ganglion (DRG) [26], thus it might be an ion channel involved in sensing mechanical pain. Piezo 1 and 2 have unique roles in mechanical transduction in sensory neurons [27,28,29]. However, there are some controversies regarding Piezo1 expression in nociceptive neurons. One group demonstrated the absence of Piezo1 in DRG neurons [15], while another presented the evidence for the expression not only of Piezo2 but also of Piezo1 in DRG neurons and found that the Piezo1 was preferentially expressed in small size DRG neurons, suggesting their role in pain [14]. Thus, channels normally expressed in specific sensory neuron are likely to play an important role in nociceptive signaling in different types of sensory neurons. 

Piezo channels are nonselective cationic ion channels that allow for the permeation of sodium, potassium, calcium, and magnesium by mechanical stimulus (stimulating cell poking, membrane stretching, substrate deflection, and fluid flow in mammalian cells) [30,31]. Activation of Piezo1 by mechanical stimuli is blocked by nonspecific inhibitors such as ruthenium red, Gd^3+^, and the spider peptide GsTMx4 (which is a more specific inhibitor of mechanosensitive cationic channels) [32,33]. Recently, Yoda1 (a small synthetic molecule) was discovered [34] as a Piezo1 activator. Yoda1 activates Piezo1 without mechanical stimuli and causes a significant change in the kinetics of the mechanical responses or sensitizing Piezo1 activation [34]. A total of 21 amino acids within a minimal Yoda1 binding motif (mouse Piezo1 residues 1961–2063), also known as the agonist transduction motif (ATM), are not conserved between Piezo1 and Piezo2. Therefore, theses amino acids may form a Piezo1-specific Yoda1 binding site or may influence Yoda1 binding to the Piezo1 pore [35]. 

Our findings demonstrated that Yoda1-induced intracellular calcium increases in mouse DRG neurons and Yoda1-induced intracellular Ca^2+^ increases disappear in the absence of extracellular Ca^2+^. This suggests that these Ca^2+^ increases are caused via activation of ion channel in plasma membranes, but not organelles in cells. These Ca^2+^ responses were inhibited by ruthenium red and GsMTx4, indicating that the channel, which was activated by Yoda1, is a mechanosensitive cation channel. However, the other mechanosensitive ion channel, TRPV4, is known to be in DRG neurons [36]. In MC353-E1 cells, TRPV4-knockdown blocked Yoda1-induced Ca^2+^ response as well as GSK1016790A, a TRPV4 agonist. Yoda1-induced Ca^2+^ response can be either in TRPV4-dependent or independent manners. When both Piezo1 and TRPV4 were highly expressed, Yoda1 induced a TRPV4-dependent Ca^2+^ response via activation of Piezo1 [37]. In addition, Dooku1, an analogue of Yoda1, modified the pyrazine ring and thiadiazole group of Yoda1, thus was effective in antagonizing Yoda1-induced Piezo1 channel activity (not constitutive Piezo1 channel activity), but it did not affect the 4α-PDD (TRPV4 agonist)-induced Ca^2+^ response [38]. In the present study, the result of Figure 3D in which Dooku1 blocks Yoda1-induced intracellular Ca^2+^ may be attributed to the activity of Piezo1 regardless of TRPV4 activity. 

Primary afferent neuron size is one of the morphological attribution of neurons and can be distributed in conduction velocity groups [39]. Small neurons give rise to C- or Aδ-fiber which are related to nociception, medium and large neurons give rise to Aα- and Aβ-fiber which are related to light touch [40,41,42], but some of medium neurons have Aδ-fiber [39]. Our results showed that Piezo1 and Piezo2 were expressed in various sized DRG neurons, but expression patterns of both were different. Piezo1 was highly expressed in medium-sized DRG neurons and Piezo2 was highly expressed in more than large-sized DRG neurons. These results demonstrate that Piezo1 and Piezo2 might be involved in transmitting pain or touch signals but might have different contributions in DRG sensory neurons.

Piezo1 has been reported to modulate peripheral signal transduction by regulating the expression and interaction of several receptors and channels [14,16]. Recent studies have shown that Piezo1 is regulated by TRPV1 in HEK293 cells cotransfected with TRPV1 and Piezo1. Capsaicin also inhibits mechanically activated (MA) currents in the DRG neuron isolated from TRPV1 reporter mice [43]. Functional interaction with the TRPV1 channels may coordinate the activity of Piezo channels and thereby influencing their various functions in specific cell types [13,43]. Activation of TRPV1 channels by capsaicin in DRG neurons blocks the activation of Piezo channels by depleting phosphatidylinositol 4,5-bisphosphate [PI(4,5)P_2_] and its precursor, phosphatidylinositol 4-phosphate [PI(4)P], through Ca^2+^-induced phospholipase C (PLC) δ activation [44], implying that Piezo channels require phosphoinositides for their activation [43]. Thus, the inhibition of Piezo channels by TRPV1 activation may partly contribute to the analgesic effect of capsaicin [44]. Our findings show the inhibition of Piezo1 function in mouse DRG neurons. We have observed that TRPV1 activation inhibits the inward current of Piezo1 by Yoda1 in whole-cell patch-clamp recordings (Figure 4). Thus, whether the inhibition of Piezo1 by TRPV1 activation in DRG neurons depends on PLC δ, β, or another signal pathway is still unclear, and the interaction of Piezo1 channels with TRPV1 requires further investigation regarding complementary or synergistic effects in DRG neurons.

In conclusion, our study demonstrates Piezo1 expression and physiological functions in mouse and human DRG neurons. Moreover, the Piezo1 channel shows coexpression with TRPV1, which can be negatively regulated by the TRPV1 channels in mouse DRG neurons. Thus, our findings suggest that the mechanosensitive Piezo1 channel should be considered as one of the molecular targets in mechanical hypersensitivity and nociception. Future studies on Piezo1 are warranted to further understand its role in mechanical touch and pain in primary sensory neurons.

## 4. Materials and Methods 

### 4.1. Reagents

Yoda1, ruthenium red, and Dooku1 were purchased from Tocris (Avonmouth, Bristol, UK). Capsaicin was purchased from Sigma (St. Louis, MO, USA) and the stock solutions were made with 99.5% ethanol. All stock solutions were stored at −20 °C. 

### 4.2. Animals

All surgical and experimental procedures were reviewed and approved by the Institutional Animal Care and Use Committee of the College of Medicine at Gachon University (approval number: LCDI-2019-0072 15 April 2019). Animal experiments were performed according to the guidelines of the International Association for the Study of Pain. Adult male C57BL/6N mice were purchased from Orientbio (Sungnam, Korea). The mice were allowed to habituate to the facility (12 h light/12 h dark cycle) for at least 1 week prior to the start of experimental procedures.

### 4.3. Mouse DRG Neuron Cultures

DRGs were removed from mice (6 to 9 weeks old) and were incubated with collagenase A (0.2 mg/mL, Roche, Basel, Switzerland)/dispase-II (2.4 units/mL, Roche) at 37 °C for 90 min. Cells were mechanically dissociated with gentle pipetting. DRG cells were plated on poly-D-lysine-coated coverslips and grown in a neurobasal culture medium with 10% fetal bovine serum (Gibco, Waltham, MA, USA), 2% B27 supplement (Invitrogen, Carlsbad, CA, USA), and 1% penicillin/streptomycin for 1 day before calcium imaging and patch-clamp recordings. 

### 4.4. Human DRG Neuron Cultures

Human DRGs were obtained from patients through the Samsung Medical Center (IRB number: 2018-11-063-002). The donor information is listed on Table 1. Postmortem DRGs were dissected from donors and delivered in an ice-cold culture medium to the laboratory at Gachon University within 24 hrs. Upon delivery, the DRGs were rapidly dissected from the nerve roots, minced in Ca^2+^-free Hank’s balanced salt solution (Gibco), and digested at 37 °C in a humidified CO_2_ incubator for 180 min with collagenase type II (Worthington, Lakewood, NJ; 390 units/mg; 12 mg/mL final concentration) and dispase II (Roche, Basel, Switzerland; 1 unit/mg, 20 mg/mL) in phosphate-buffered saline (PBS) with 200 μM sodium pyruvate and 10 mM 4-(2-hydroxyethyl)-1-piperazineethanesulfonic acid (HEPES). The pH was adjusted to 7.4 with NaOH. The sample was then centrifuged for 5 min at 400× *g* to pellet the ganglia and the collagenase type II and dispase II solution was carefully removed. Five mL of prewarmed Dulbecco’s modified Eagle medium (DMEM) supplemented with 10% FBS and 1% penicillin/streptomycin was added, and the cells were mechanically dissociated. The solution was then filtered through a 100-μm nylon mesh and centrifuged for 5 min (500× *g*). The DRG cell pellet was resuspended and plated on 0.1 mg/mL Corning^®^ Cell-Tak-coated glass coverslips. The DRG cultures were grown in neurobasal medium supplemented with 10% FBS, 2% B-27 supplement, 1% N_2_ supplement, and 1% penicillin/streptomycin. After five days of culture, human DRG neurons were used for up to four weeks. 

### 4.5. shRNA Downregulation Assay and Taqman Assay

Primary cultured DRG neurons were transfected with short-hairpin RNA (shRNA) 1 h after plating. One day after infection, the media was changed and the cells were incubated for 4 days. A TaqMan assay and calcium imaging were then performed. For the TaqMan assay, primary cultured DRG neurons were isolated rapidly. Total RNAs were extracted according to the manufacturer’s instructions. RNAs (0.5–1 μg) were reverse-transcribed using M-MLV (Invitrogen). The expression level of *piezo1* was measured by qPCR using the TaqMan^®^ assay (Thermo Fisher Scientific, Waltham, MA, USA). The PCR reaction was performed two times with quadruplicate or quintuplicate containing 2× TaqMan gene expression master mix, 20× TaqMan gene expression assay, and cDNA (final volume of 20 μL). After the incubation step at 50 °C for 2 min, the initial denaturation step at 95 °C for 10 min was followed by 40 cycles at 95 °C for 15 s, and 60 °C for 1 min. 2^-ΔΔCt^ values were analyzed with *gapdh* as a reference gene.

### 4.6. Calcium Imaging

Neurons on poly-D-lysine-coated coverslips were loaded with 2 μM Fura-2 AM (Thermofisher, Massachusetts) for 40 min at 37 °C in DMEM and then transferred to the chamber. They were then placed onto the inverted microscope (Olympus BX51WI, Tokyo, Japan) and perfused continuously with a balanced bath solution containing 140 mM NaCl, 5 mM KCl, 2 mM CaCl_2_, 1 mM MgCl_2_, 10 mM HEPES, and 10 mM glucose. The solution was adjusted to a pH of 7.4 with NaOH. Calcium imaging was conducted at 25 °C. Cells were illuminated with a 175-watt xenon arc lamp, and excitation wavelengths (340/380 nm) were selected by a Lambda DG-4 monochromator wavelength changer (Shutter Instrument, Novato, CA, USA). The fluorescence 340/480 nm ratio was measured by digital video microfluorometry with an intensified camera (optiMOS, QImaging, Surrey, BC, USA) coupled to a microscope and software (Slidebook 6, 3i, Intelligent Imaging Innovations, Denver, CO, USA). The perfusion system was driven by gravity and a flow speed of 1 mL/min. Yoda1 and capsaicin were treated every 5 min. In the experiment using antagonists, we applied the first application of Yoda1 for 10 s and then washed the cells using a solution containing 2Ca^2+^. The cells were then pretreated with the antagonists for 1 min before the solution containing antagonists and the second application of Yoda1 was immediately applied for 10 s. In the calcium experiment, a third application of Yoda1 was applied for 10 s to confirm the recovery of Piezo1 activation. Finally, neurons were identified through their reaction with a high concentration of KCl as a neuronal marker.

### 4.7. Whole-Cell Patch-Clamp Recordings 

Whole-cell patch-clamp recordings were conducted at 25 °C using MPC-200 manipulators (Sutter Instrument) and a Multiclamp 700B amplifier (Molecular Devices, San Jose, CA, USA). The patch pipettes were pulled from borosilicate capillaries (Chase Scientific Glass Inc., Rockwood, TN). The resistance of the pipette was 2–5 megohms, and the series resistance was compensated (> 80%). Data were low-pass filtered at 2 kHz and sampled at 10 kHz. Voltage clamp recordings were performed at a holding potential of −70 mV. The pipette solution for voltage clamp experiments contained, 122.5 mM K-gluconate, 12.5 mM KCl, 0.2 mM EGTA, 8 mM NaCl, 2 mM MgATP, 0.3 mM Na_3_GTP, and 10 mM HEPES; the pH was adjusted to 7.3 with KOH. The bath solution contained, 119 mM NaCl, 2.5 mM KCl, 26 mM NaHCO, 2.5 mM CaCl_2_, 1.3 mM MgSO_4_, and 25 mM glucose; the pH was 7.4. 

### 4.8. Single-Cell RT-PCR (scRT-PCR)

The entire single-cell solution was aspirated into a pipette under visual control via negative pressure. DRG neurons were visualized with a 20× water-immersion objective using a microscope and diameters that were measured using Slidebook 6 software. Pipettes used for the entire neuron harvest had a tip diameter range of about 20 µm and were filled with RNase OUT (invitrogen) and diethylpyrocarbonate (DEPC) treated water. The tip of the pipette and its contents were broken into a reaction tube containing reverse transcription (RT) reagents. RT was carried out for 60 min at 50 °C (superscript III, Invitrogen). cDNA was used in separate PCRs. All PCR amplifications were performed with nested primers (Table 2). The first round of PCR was performed in 25 μL of PCR buffer containing 0.4 mM dNTP mix, 0.4 μM “outer” primers, 1 μL of RT product, and 0.1 μL of Platinum Taq DNA polymerase (invitrogen). The protocol included 5 min of initial denaturation at 94 °C, followed by 35 cycles of 30 s of denaturation at 94 °C, 30 s of annealing at 57 °C, and 30 s of elongation at 72 °C, and was completed with 10 min of final elongation. For the second round of amplification, PCR premix with “inner” primers and 1–5 μL of the products from the first round of PCR and DEPC-DW (up to 20 μL) were used. The reaction was the same as the first round, except for 20 cycles of 30 s of denaturation. For the positive controls, *gapdh* or *neun* primers were used in parallel for each PCR reaction. The PCR products were then displayed on ethidium bromide-stained 1.2% agarose gel. Gels were photographed using a GelDoc System (Bio-rad, Hercules, CA, USA).

### 4.9. Western Blot

Mice were perfused with PBS, and tissues were collected. Tissues were homogenized in a lysis buffer containing a protease inhibitor cocktail. The protein samples were loaded for each lane and separated by a 6% SDS-PAGE gel. After transferring to a polyvinylidene fluoride (PVDF) membrane, the blots were incubated overnight at 4 °C with a rabbit polyclonal antibody against Piezo1 (1:500, Proteintech) and a mouse monoclonal antibody against β-actin (1:1000, Sigma). These blots were incubated further with a horseradish peroxidase (HRP)-conjugated secondary antibody and developed in an enhanced luminol-based chemiluminescent substrate (ECL) solution. Specific bands were evaluated by apparent molecular sizes.

### 4.10. Statistical Analysis

All data are expressed as mean ± standard error of the mean. Biochemical, electrophysiological, and calcium imaging data were analyzed using Student’s t-test (two groups) and one-way or two-way ANOVA followed by post hoc Fisher’s LSD test using Prism 6 software. *p*-values less than 0.05 were considered statistically significant.

## Figures and Tables

**Figure 1 ijms-21-03834-f001:**
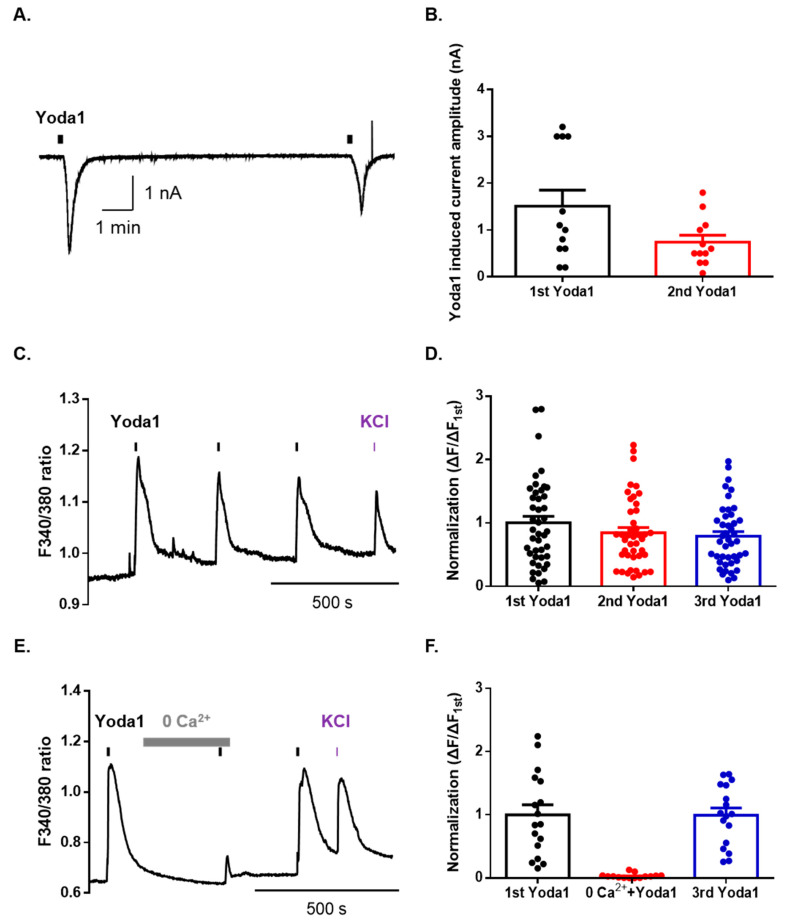
Yoda1 induces extracellular cation influx via Piezo1 in mouse dorsal root ganglion (DRG) sensory neurons. (**A**) A representative trace of Yoda1 (10 μM, 10 s) induced an inward current in DRG neurons at a holding potential of −60 mV. (**B**) The mean peak amplitude of Yoda1 (10 μM, 10 s) induced an inward current in mouse DRG neurons (*n* = 12). (**C**) Ca^2+^ response induced by sequential application of Yoda1 (10 μM, 10 s) in mouse DRG neurons. (**D**) Mean normalized amplitude of sequential Yoda1-induced Ca^2+^ response (*n* = 43). (**E**) The Ca^2+^ response in the presence or absence of extracellular Ca^2+^. Bar (gray) indicates the 0 mM CaCl_2_ extracellular solution applied to mouse DRG neurons. (**F**) Mean normalized amplitude of sequential Yoda1-induced Ca^2+^ response (*n* = 17). High potassium chloride (50 mM) is used as the neuronal marker. All results are presented as the mean ± standard error of the mean (SEM).

**Figure 2 ijms-21-03834-f002:**
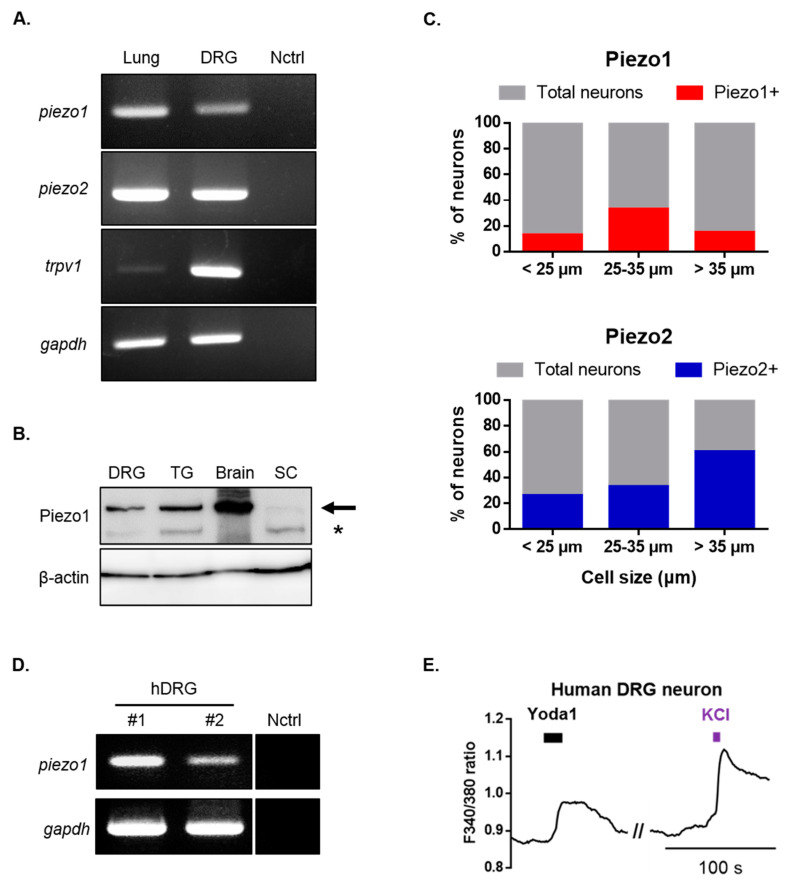
Identification of Piezo1 expression in mouse and human dorsal root ganglion (DRG) neurons. (**A**) RT-PCR showing *p**iezo1, piezo2, trpv1 and gapdh* mRNA expression in mouse DRG and lung tissue (*piezo1* and *piezo2* positive control). Primer and ultrapure water were used as negative controls. (**B**) Western blot showing Piezo1 protein expression in mouse DRG, trigeminal ganglion (TG), whole brain, and spinal cord (SC). The upper bands which are indicated by arrow represent Piezo1. Second bands (*) is nonspecific. (**C**) Single-cell RT-PCR of mouse DRG neurons showing % of *piezo1* and *piezo2* mRNA expressing neurons. (**D**) RT-PCR showing *piezo1* gene expression in two different human’s DRG (human primer and ultrapure water were used as negative controls). (**E**) Application of Yoda1 (10 μM, 10 s)-induced Ca^2+^ response in human DRG neurons. High potassium chloride (50 mM) is a neuronal marker. DRG: dorsal root ganglion; Nctrl: negative control; RT-PCR: reverse transcription-polymerase chain reaction; SC: spinal cord; TG: trigeminal ganglion; hDRG: human DRG.

**Figure 3 ijms-21-03834-f003:**
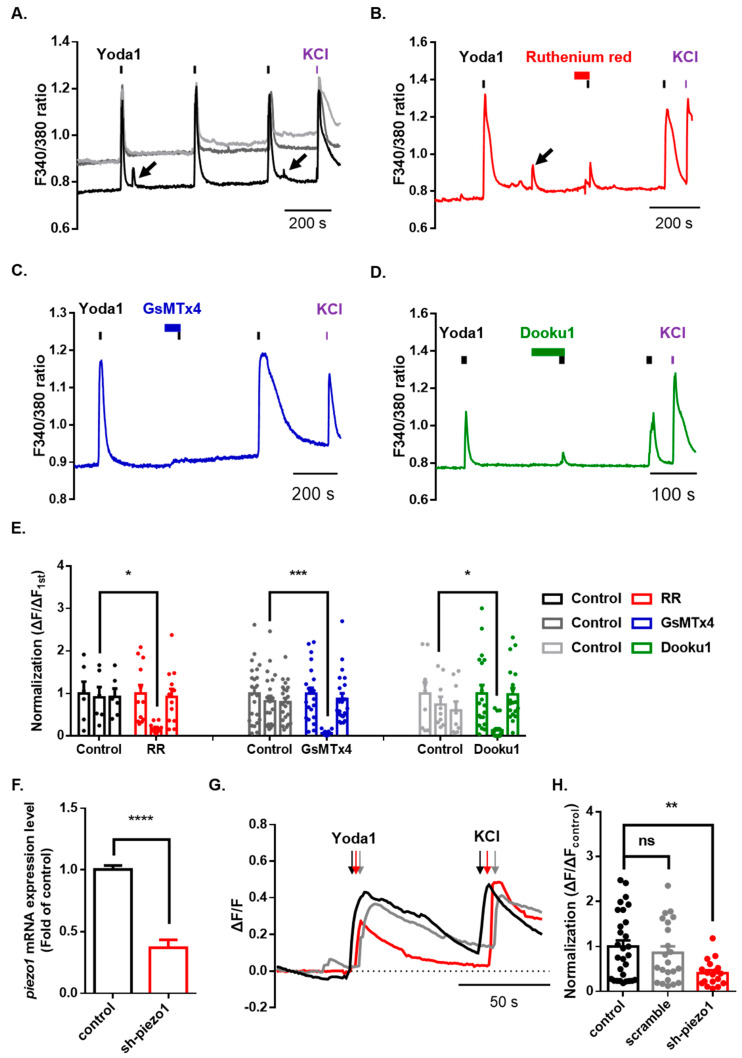
Inhibition of Piezo1 by mechanical-ion channel inhibitors and short hairpin RNA-mediated silencing of Piezo1. (**A**) The Yoda1 (10 μM, 10 s)-induced intracellular Ca^2+^ increase. The Yoda1 (10 μM, 10 s)-induced intracellular Ca^2+^ increase is reversibly inhibited by ruthenium red (30 μM, N and P-type nonselective Ca^2+^ channel blocker, red) (**B**), GsMTx4 (2.5 μM, selective stretch-activated channel blocker, blue) (**C**), and Dooku1 (10 μM, Yoda1 analogue, green) (**D**). The arrows indicate spontaneous intracellular Ca^2+^ increase. (**E**) The bars indicate treatment with various inhibitors: ruthenium red (RR) (control, *n* = 6; RR, *n* = 12, two-way ANOVA (Fisher’s LSD test), *, *p* < 0.05); GsMTx4 (control, *n* = 23; Gs, *n* = 22, two-way ANOVA (Fisher’s LSD test), ***, *p* < 0.001); and Dooku1 (control, *n* = 9; Dk, *n* = 19, two-way ANOVA (Fisher’s LSD test), *, *p* < 0.05). (**F**) *piezo1* mRNA levels, normalized by *gapdh,* were quantified by a TaqMan assay after transfection with sh-piezo1 dorsal root ganglion (DRG) neurons (unpaired t-test, **** *p* < 0.0001). (**G**) Application of Yoda1 (10 μM, 10 s)-induced Ca^2+^ response in mouse DRG neurons transfected with sh-piezo1 (red) and scramble (gray) or control (black). (**H**) Mean normalized amplitude of Yoda1-induced Ca^2+^ response (control, *n* = 29; scramble, *n* = 22; sh-piezo1, *n* = 18, one-way ANOVA (Dunn’s multiple comparisons test, ns; nonsignificant, **, *p* < 0.01). High potassium chloride (50 mM) is a neuronal marker. All results are presented as the mean ± standard error of the mean (SEM) (control versus sh-piezo1 and scramble or control versus each drug). DRG: dorsal root ganglion; sh: short hairpin.

**Figure 4 ijms-21-03834-f004:**
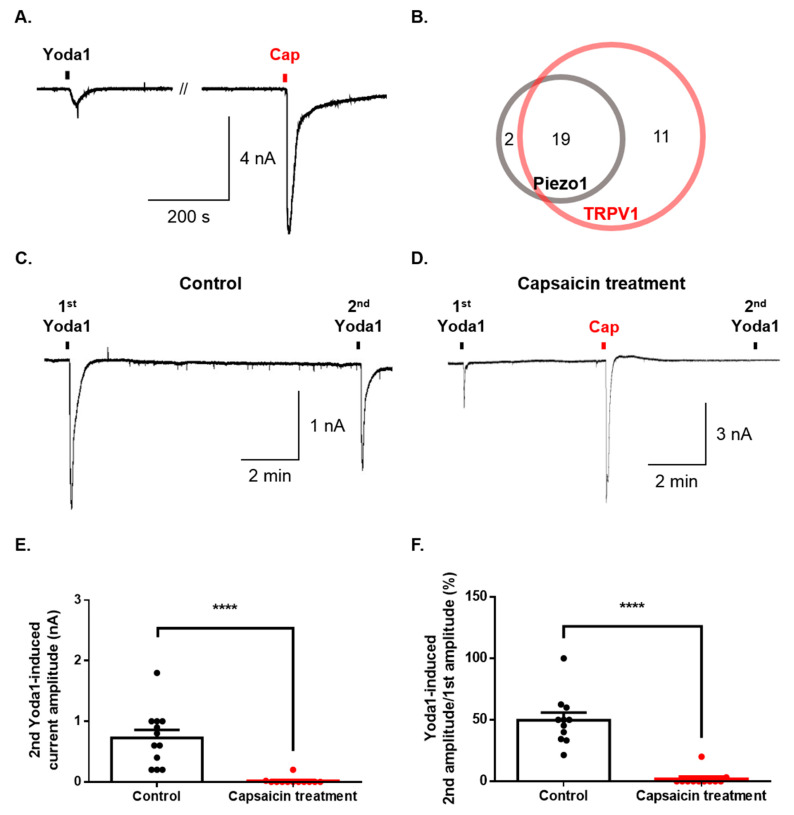
Regulation of Piezo1 by TRPV1 activation in the same dorsal root ganglion (DRG) neurons. (**A**) Representative trace of the inward current induced by Yoda1 (10 μΜ, 10 s) and capsaicin (cap, 100 nM, 10 s) in the same neuron. (**B**) The number of neurons activated by Yoda1 or/and capsaicin. (**C**) A representative trace of the Yoda1 (10 μM, 10 s)-induced inward current (control group). (**D**) A representative trace of the effect of the inward current of Piezo1 by capsaicin (cap, 100 nM, 10 s) (capsaicin treatment group). (**E**) The mean peak amplitude of the second application of Yoda1 (10 μM, 10 s) after capsaicin treatment (**D**) or not (**C**) (*n* = 12, unpaired t-test, ****, *p* < 0.0001). (**F**) The percentage of the second Yoda1-induced amplitude after capsaicin treatment (or not) compared to the first Yoda1-induced amplitude (**C**) and (**D**) (*n* = 11, unpaired t-test, ****, *p* < 0.0001). The results are presented as the mean ± standard error of the mean (SEM).

**Table 1 ijms-21-03834-t001:** Human DRG donor information.

Sex	Age (Years)
Female	73
Male	40
Female	64
Female	60

**Table 2 ijms-21-03834-t002:** Primer information for standard RT-PCR and scRT-PCR amplification.

Target Gene (Product Length)		Forward (5′-3′)	Reverse (5′-3′)
**mouse *piezo1***	Outer (239 bp)	TCCCAGAAGATGAGATGGCA	ACCCACATAAAGCTGGTCCA
	Inner (176 bp)	CCGTAGCCACATGATGCAG	TCACCCGAAGAAGCTCCTG
**mouse *piezo2***	Outer (418 bp)	TGGACAGCGAATGGACTTCT	CCTCGTTCAGCCAGCATAAC
	Inner (222 bp)	TGATTCATGCCTGTTGGTTG	TGAAATCCGGGAAGTACAGC
**mouse *trpv1***	273 bp	TGATCATCTTCACCACGGCTG	CCTTGCGATGGCTGAAGTACA
**mouse *gapdh***	Outer (367 bp)	AGCCTCGTCCCGTAGACAAAA	TTTTGGCTCCACCCCTTCA
	Inner (313 bp)	TGAAGGTCGGTGTGAACGAATT	GCTTTCTCCATGGTGGTGAAGA
**mouse *neun***	Outer (473 bp)	CCAAGGGTTTTGGGTTTGTA	ACAAGAGAGTGGTGGGAACG
	Inner (202 bp)	CCAAGGGTTTTGGGTTTGTA	TCAGGCCCATAGACTGTTCC
**human *piezo1***	182 bp	AGATCTCGCACTCCAT	CTCCTTCTCACGAGTCC
**human *gapdh***	459 bp	CAAATTCCATGGCACCGTCA	ATGATGTTCTGGAGAGCCCC

RT-PCR: reverse transcription-polymerase chain reaction; scRT-PCR: single-cell reverse transcription-polymerase chain reaction.

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
