# Peer review of "Functional Expression of Piezo1 in Dorsal Root Ganglion (DRG) Neurons"

_ijms, 2020, doi:10.3390/ijms21113834_

Round 1
Reviewer 1 Report
We appreciate the effort of the authors to improve the data presentation and the description of the methodologies used.
However, an important point remains unresolved: the difference between whole cell and fura-2 results (see figure 4A and 4F). The patch-clamp technique in whole-cell configuration produces cell dialysis and is quite invasive. It would be important to understand whether the effect of capsaicin activation of TRPV1 on the currents activated by Yoda1 is still observed through less invasive techniques. On the other hand, calcium dyes too can interfere with the intracellular calcium homeostasis and therefore alter the intracellular signaling. To overcome this point, I suggest performing the stimulation protocol shown in Figures 4A and 4F by means of extracellular recording techniques of the electrical activity of neurons. The best way would be to use a Multi Electrode Array system. Alternatively, the authors can record the electrical activity evoked by the agonists in the cell-attached configuration with the pipette filled with extracellular solution.
Author Response
# Reviewer 1
Q1. We appreciate the effort of the authors to improve the data presentation and the description of the methodologies used.
However, an important point remains unresolved: the difference between whole cell and fura-2 results (see figure 4A and 4F). The patch-clamp technique in whole-cell configuration produces cell dialysis and is quite invasive. It would be important to understand whether the effect of capsaicin activation of TRPV1 on the currents activated by Yoda1 is still observed through less invasive techniques. On the other hand, calcium dyes too can interfere with the intracellular calcium homeostasis and therefore alter the intracellular signaling. To overcome this point, I suggest performing the stimulation protocol shown in Figures 4A and 4F by means of extracellular recording techniques of the electrical activity of neurons. The best way would be to use a Multi Electrode Array system. Alternatively, the authors can record the electrical activity evoked by the agonists in the cell-attached configuration with the pipette filled with extracellular solution.
Response 1: Thank you for your important comment. We tried to perform the cell attached patch clamp with the pipette filled with extracellular solution (with Yoda1). However, we did not record the electrical activity caused by the agonist using cell attached configuration since it is difficult to record single or multi- channel of Piezo1. We had a similarly experience with an experiment using whole cell patch clamp to record inward current by Yoda1; but it is not easy to record inward current by yoda1. When successfully recording inward current by yoda1, the amplitude of current varies from 0.5 nA to 2~3nA. The probability of successful recording is relatively low for various reasons. For this reason, it was currently inevitable to remove the calcium data in Figure 4, and we reorganized the results with the exception of the calcium results again since it is difficult to prove different results between patch clamp and calcium imaging data. So, if we had more time, we would have tried; and in the future, we intend to conduct the relevant experiments to obtain such results. Thanks again for the help of the reviewer.
Reviewer 2 Report
Review-ijms-787241
The study entitled “Functional Expression of Piezo1 in Dorsal Root Ganglion (DRG) Neurons” by Roh et al., proposes through Ca2+ imaging and patch-clamp experiments on mammalian DRGs that the Piezo1 channel, which is known to be a cation channel activated by mechanical stimulation, is expressed there and pharmacologically activated by the Yoda1 molecule.
I have read this publication carefully, as well as the responses to the comments of the 2 reviewers who had reviewed this publication.
Sometimes the conclusions drawn from experiments seem inappropriate or hasty. There are quite a few semantic shortcuts in the draft. I ask the authors not to over-interpret their data. They have interesting results but which cannot allow them to anticipate a precise function of Piezo1 in the DRGs.
From a pharmacological point of view, what may seem very limiting is the use of molecules supposed to act on Piezo1, but without the specificity of these molecules being known. Thus, Yoda1 has been described as a specific activator of Piezo1, but it is not known whether it is capable of acting on another receptor. Likewise, it is well known that ruthenium red or GsMTx4 are inhibitors of Piezo channels, but without specificity.
Finally, the discussion, even reworked, is not of a good level. It is often off-topic, and must be refocused on the subject of this study. You have to rewrite it, and restructure it.
L19-20: The abstract states that “Piezo channels are mechanosensitive ion channels in the somatosensory system. Piezo1 is primarily expressed in non-sensory tissues”. This is contradictory.
“Piezo channels are mechanosensitive ion channels. Piezo1 is primarily…”
L20-22: “Piezo1 is primarily expressed in non-sensory tissues [...] However, the physiological functions of Piezo1 in DRG neurons remain poorly understood”. Again this is contradictory: if Piezo1 is expressed in non-sensory tissues, why would it be present in DRGs? Especially since this study shows (for the first time?) its expression in DRGs. To rephrase.
L38: “several transduction-mediating molecules have been identified” is not clear. Did you mean “the pharmacology aspects of their transduction into electrical messages have been identified”?
L55-56: “These results indicate that there may be other channels in the DRG that may affect the function of mechanical pain”. A recent study identifies the ion channel TACAN as playing this role. Its neither mentioned nor discussed here (Beaulieu-Laroche et al., 2020. doi: 10.1016/j.cell.2020.01.033). This channel could fit to the function you are mentioning (see L194-195).
L61: “It is important to address the expression and function of Piezo1 activity in DRG neurons…”
L70-71: Functionally, Yoda1 induces an increase in intracellular calcium and an inward current through the activation of Piezo1 in small and medium-sized DRG neurons.
L80-81: “small-sized neurons, 50% (10/20) were medium-sized neurons, and 13% (2/15) were large-sized neurons”. Please give at lines 350-351 the exact diameter range (in µm) of small-, medium- and large-sized neurons. I guess it corresponds to <25 µm (small), 25-25 µm (medium) and >35 µm (large) as indicated in Figure 2C but it is not indicated.
L86-87: Extracellular Ca2+ removal abolished Yoda1-induced intracellular Ca2+ increase, in a reversible manner (Figure 1EF).
L87-88: Therefore, these results suggest that Piezo1 can be pharmacologically activated in mouse DRG neurons, and drives a cation influx.
Figure 1: The current trace in Fig. 1A shows an amplitude of about 5-6 nA after Yoda1 addition. But the distribution of current amplitudes in Fig. 1B shows amplitudes between 0.2 and 3 nA. How do you explain this discrepancy? Is the figure incomplete? Or the scale bar wrong? Please correct.
Figure 2 shows the expression pattern of Piezo1 as a gene and as a protein in mouse and human DRGs. The question I have towards this result is whether it is co-expressed with Piezo2 or not? Why did the authors not check the presence of Piezo2 at the same time? It looks important to understand the putative interplay of both channels, especially in cells where Piezo is expected.
Figure 2B: Two bands can be seen on the WB (Piezo1). What does the second one correspond to ? Please precise. Is it a non-specific one?
L125: various cation channel inhibitors
L126-127: Ruthenium red in not a mechanically-activated current blocker! RR is totally non-specific, and cannot be used to target solely Piezo channels. It is used to block various TRP channels, notably TRPV1. GsMTx4 has been shown to act on different channels (TRPV4, TRPA1, TRPC1, TRPC6) and is not specific of Piezo1. These elements should be brought to the reader and need to be addressed as a subject for discussion, if not for debate.
Figure 3: Several unclear elements in this figure.
Fig 3A: What does the second (and small) peak correspond to? Nothing above the curve indicates it.
Fig 3ABC: On the right, it is not clear to what the 6 histograms refer to. I understand that you stimulated DRGs 3 times with Yoda1. The first application is Yoda1 alone, the second Yoda1 + RR/GsMTx4/Dooku1, the third Yoda1 alone. That should make 3 histograms, not 6. Please revise Figure 3 as it is not clear to read.
Figure 4 intends to show that Piezo1 and TRPV1 are co-expressed in a subset of DRGs. This is done with Ca2+ imaging and patch-clamp. But wouldn't the simplest have been simply to make a WB as in figure 2B? Why didn’t the authors perform this straightforward experiment? I think it is essential since the results in Figure 4 are not clear (Fig 4A and 4F are contradictory).
Fig 4GH is not clear at all.
L183: “suggesting that Piezo1 plays a significant part in the mechanosensory function”. How can you say that? What in your experiments let you think that Piezo1 could play a significant part of such function, in the absence of any mechanical stimulation?
L183-184: “reveal that Piezo1 is expressed in DRG neurons of various sizes”
L193: “TRPV1, which is one of the channels involved in heat-nociception”. The same has been written in L188-189.
L199: Piezo1 is blocked by non-specific inhibitors such as…
L215-224 and 245-257: I do not understand the signification of these paragraph sin the discussion. They do not deal with the subject of this article.
L258-259: The sentence “we have confirmed the physiological function of Piezo1 expression” is wrong, since no evidence on the role of Piezo1 is brought by this article.
L265: with TRPV1, which might regulate Piezo1 functions
L266-267: “Piezo1 channel should be considered as one of the molecular targets in mechanical hypersensitivity and nociception”. How can it be stated? The authors should be aware of the fact that concerning nociceptors, especially unmyelinated DRG neurons:
- Most small-diameter C fibers include nociceptors which are both heat and mechanically sensitive (CMHs).
- Other C fibres include heat-responsive, but mechanically insensitive “silent” nociceptors. They develop a mechanical sensitivity only in the case of injury…
One of them might correspond to the TRPV1- and Piezo1-expressing cells.
Even reworked, the discussion still needs to be ameliorated to discuss the data generated as part of this study, which is essentially a pharmacological study, not a physiological study. Indeed, nothing is tackled, experimentally speaking, on the function of Piezo1.
Minor
L43: “of the Piezo channel family”
Figure 1F: 0Ca2+ + Yoda1
L79: in mouse DRG neurons (Figure 1A and B)
L123: 2.3. Pharmacological Inhibition of Piezo1 in Mouse DRG Neurons
L141: The Yoda1 (10 μM, 10 s)-induced intracellular Ca2+ increase
L162: Yoda1-induced Ca2+ responses
L204: Thus, Yoda1 selectively activates Piezo1 but not Piezo2
Author Response
# Reviewer 2
From a pharmacological point of view, what may seem very limiting is the use of molecules supposed to act on Piezo1, but without the specificity of these molecules being known. Thus, Yoda1 has been described as a specific activator of Piezo1, but it is not known whether it is capable of acting on another receptor. Likewise, it is well known that ruthenium red or GsMTx4 are inhibitors of Piezo channels, but without specificity.
Q1
L19-20: The abstract states that “Piezo channels are mechanosensitive ion channels in the somatosensory system. Piezo1 is primarily expressed in non-sensory tissues”. This is contradictory.
“Piezo channels are mechanosensitive ion channels. Piezo1 is primarily…”
L20-22: “Piezo1 is primarily expressed in non-sensory tissues [...] However, the physiological functions of Piezo1 in DRG neurons remain poorly understood”. Again this is contradictory: if Piezo1 is expressed in non-sensory tissues, why would it be present in DRGs? Especially since this study shows (for the first time?) its expression in DRGs. To rephrase.
Response 1: Thank you for your comment and suggestion. We have revised the abstract accordingly.
L19-21: “: Piezo channels are mechanosensitive ion channels. Piezo1 is primarily expressed in non-sensory tissues, whereas Piezo2 is predominantly found in sensory tissues, including dorsal root ganglion (DRG) neurons.”
L21-22: “However, a recent study demonstrated the intracellular calcium response to Yoda1, a selective Piezo1 agonist, in TG neurons.”
Q2
L38: “several transduction-mediating molecules have been identified” is not clear. Did you mean “the pharmacology aspects of their transduction into electrical messages have been identified”?
Response 2: We have revised the text to:
L37: "Chemical and thermal-induced nociceptor responses in sensory neurons has been well studied"
Q3
L55-56: “These results indicate that there may be other channels in the DRG that may affect the function of mechanical pain”. A recent study identifies the ion channel TACAN as playing this role. Its neither mentioned nor discussed here (Beaulieu-Laroche et al., 2020. doi: 10.1016/j.cell.2020.01.033). This channel could fit to the function you are mentioning (see L194-195).
Response 3: Thank you for your comment. We have added a paragraph with TACAN information in discussion.
L187-189: “It has recently been found that TACAN share no sequence similarity with known Piezos and Transmembrane channels like (TMC) protein [26]. TACAN is expressed predominately in small-diameter neuron of DRG [26], thus it might be an ion channel involving in sensing mechanical pain.”
Q4
L61: “It is important to address the expression and function of Piezo1 activity in DRG neurons…”
Response 4: We revised the sentence to:
L60: “It is important to address the expression and function of Piezo1 activity in DRG neurons…”
Q5
L70-71: Functionally, Yoda1 induces an increase in intracellular calcium and an inward current through the activation of Piezo1 in small and medium-sized DRG neurons.
Response 5: We revised the text to:
L70-71: "Functionally, Yoda1 induces an increase in intracellular calcium and an inward current through the activation of Piezo1 in small and medium-sized DRG neurons."
Q6
L80-81: “small-sized neurons, 50% (10/20) were medium-sized neurons, and 13% (2/15) were large-sized neurons”. Please give at lines 350-351 the exact diameter range (in µm) of small-, medium- and large-sized neurons. I guess it corresponds to <25 µm (small), 25-25 µm (medium) and >35 µm (large) as indicated in Figure 2C but it is not indicated.
Response 6: We have added size units in the result section 2.1
L80-82: “…small (<25 μm)-sized neurons, 50% (10/20) were medium (25-35 μm)-sized neurons, and 13% (2/15) were large (>35 μm)-sized neurons.”
Q7
L86-87: Extracellular Ca2+ removal abolished Yoda1-induced intracellular Ca2+ increase, in a reversible manner (Figure 1EF).
Response 7: We have revised the text to:
L86-87: “Extracellular Ca2+ removal abolished Yoda1-induced intracellular Ca2+ increase, in a reversible manner (Figure 1E and F).”
Q8
L87-88: Therefore, these results suggest that Piezo1 can be pharmacologically activated in mouse DRG neurons, and drives a cation influx.
Response 8: We revised the text to:
L87-88: “Therefore, these results suggest that Piezo1 can be pharmacologically activated in mouse DRG neurons and drives a cation influx."
Q9
Figure 1: The current trace in Fig. 1A shows an amplitude of about 5-6 nA after Yoda1 addition. But the distribution of current amplitudes in Fig. 1B shows amplitudes between 0.2 and 3 nA. How do you explain this discrepancy? Is the figure incomplete? Or the scale bar wrong? Please correct.
Response 9: Thank you for your comment and question and we apologize for the lack of clarity. We changed the scale bar in Figure 1A.
L89: Figure 1A
Q10
Figure 2 shows the expression pattern of Piezo1 as a gene and as a protein in mouse and human DRGs. The question I have towards this result is whether it is co-expressed with Piezo2 or not? Why did the authors not check the presence of Piezo2 at the same time? It looks important to understand the putative interplay of both channels, especially in cells where Piezo is expected.
Response 10: Thank you for your comment and we agree with your assessment. When we started this study, we assessed the gene expression of some mechanical stimuli sensing channels including piezo2 as well as piezo1 in mouse DRGs (data not shown in manuscript). The fact that Piezo2 is expressed in DRG is already known, hence we overlooked adding data regarding Piezo2. Fortunately, we had some single DRG neuron cDNA for identifying Piezo2 gene expression. We have confirmed that Piezo2 gene is expressed in DRG neurons of all sizes and found that the ratio of expression in large-sized neurons (12/20, 60%) was higher than that of small-sized neurons (7/27, 26%) and medium-sized neurons (3/9, 33%). Please understand that due to time constraint and high cost, it is difficult to confirm Piezo2 protein expression.
L108-110: “Piezo2 mRNA was detected in 39% of all DRG neurons (22/56), of which 26% (7/27), 33% (3/9) and 60% (12/20) were small-, medium-, and large-sized DRG neurons.”
L114: Figure 2A and C
L115-117: RT-PCR showing piezo1, piezo2, trpv1 and gapdh mRNA expression in mouse DRG and lung tissue (piezo1 and piezo2 positive control).
L119-120: “(C) Single-cell RT-PCR of mouse DRG neurons showing % of piezo1 and piezo2 mRNA expressing neurons.”
Q11
Figure 2B: Two bands can be seen on the WB (Piezo1). What does the second one correspond to ? Please precise. Is it a non-specific one?
Response 11: Thank you for your question and we apologize for the lack of clarity. We think that the second band is non-specific. We thought carefully about what second band could be and what band is indeed Piezo1. The manufacturer (Proteintech) indicated that the calculated molecular weight of Piezo1 antibody (CAT No. 15929-1-AP) is 286 kDa, but it can be also observed at 233-286 kDa. Then we looked for another study that used the antibody we found similar bands like ours. The authors treated piezo1 siRNA, only upper band intensity was decreased and described the second band as “non-specific” (Nature. 2014 Nov 13; 515(7526): 279–282.). In addition, we also tested another manufacturer’s antibody which correspond to different amino acid sequence. The latter antibody also mainly detected band greater than 250 kDa of marker size (data not shown in manuscript) in DRGs. We added indicators (arrow and *) and a comment that the second band is non-specific into figure 2 legend.
L114: Figure 2B
L118-119: “The upper bands which are indicated by arrow represent Piezo1. Second bands (*) is non-specific.”
Q12
L125: various cation channel inhibitors
Response 12: Thank you for your suggestion, we revised the text accordingly.
L128: “various cation channel inhibitors.”
Q13
L126-127: Ruthenium red in not a mechanically-activated current blocker! RR is totally non-specific, and cannot be used to target solely Piezo channels. It is used to block various TRP channels, notably TRPV1. GsMTx4 has been shown to act on different channels (TRPV4, TRPA1, TRPC1, TRPC6) and is not specific of Piezo1. These elements should be brought to the reader and need to be addressed as a subject for discussion, if not for debate.
Response 13: Thank you for your comment. We have revised the text accordingly. First, we used Ruthenium red to show that Yoda1 receptor is cationic channel, then used GsMTx4 to show that Yoda1 receptor is cationic mechanosensitive channel.
L129-130: “. Ruthenium red (RR), a non-selective cation channel blocker, significantly inhibited Yoda1-induced Ca2+ increase…”
L131-133: “we used spider venom peptide that inhibits cationic mechanosensitive channels, GsMTx4, which almost completely blocked Yoda1-induced Ca2+ increase…”
Q14
Figure 3: Several unclear elements in this figure.
Fig 3A: What does the second (and small) peak correspond to? Nothing above the curve indicates it.
Response 14: The peak is spontaneous calcium transient. Sometimes, some neurons are activated in bath solution without applying drugs (Figure 3C in Physiol. Res. 66: 425-439, 2017). We denoted using arrows in Figure 3A and B and further described what it is in the figure legend.
L143: Figure 3A-E
L148-149: “The arrows indicate spontaneous intracellular Ca2+ increase.”
Q15
Fig 3ABC: On the right, it is not clear to what the 6 histograms refer to. I understand that you stimulated DRGs 3 times with Yoda1. The first application is Yoda1 alone, the second Yoda1 + RR/GsMTx4/Dooku1, the third Yoda1 alone. That should make 3 histograms, not 6. Please revise Figure 3 as it is not clear to read.
Response 15: We have revised the histograms in Figure 3. We have represented control traces in Figure 3A, RR/GsMTx4/Dooku1 treated trace in Figure 3B,C and D and normalized histograms in Figure 3E. Also we have revised the legends.
L143: Figure 3A-E
L145-152: “(A) Yoda1 (10 μM, 10 s)-induced intracellular Ca2+ increase. The Yoda1 (10 μM, 10 s)-induced intracellular Ca2+ increase is reversibly inhibited by Ruthenium red (30 μM, N and P-type nonselective Ca2+ channel blocker, red) (B), GsMTx4 (2.5 μM, selective stretch-activated channel blocker, blue) (C), and Dooku1 (10 μM, Yoda1 analogue, green) (D). The arrows indicate spontaneous intracellular Ca2+ increase. (E) The bars indicate treatment with various inhibitors.…”
Q16
Figure 4 intends to show that Piezo1 and TRPV1 are co-expressed in a subset of DRGs. This is done with Ca2+ imaging and patch-clamp. But wouldn't the simplest have been simply to make a WB as in figure 2B? Why didn’t the authors perform this straightforward experiment? I think it is essential since the results in Figure 4 are not clear (Fig 4A and 4F are contradictory).
Response 16: Figure 4 intends to show that the both channels are co-expressed in DRG single neuron and have actually functioned by agonists, Yoda1 and capsaicin. We have conducted RT-PCR and identified piezo1 and trpv1 mRNA are expressed in DRGs.
L114: Figure 2A
L102-103: “…trpv1 mRNA were expressed in DRG (Figure 2A).”
Q17
Fig 4GH is not clear at all.
Response 17: Thank you for your comment and we apologize for the lack of clarity. We have added both index and notation in Figure 4C and D and edited X and Y axis of Figure 4E and F.
L169: Figure 4C-F (We have deleted calcium imaging data and changed figure arrangement)
Q18
L183: “suggesting that Piezo1 plays a significant part in the mechanosensory function”. How can you say that? What in your experiments let you think that Piezo1 could play a significant part of such function, in the absence of any mechanical stimulation?
L183-184: “reveal that Piezo1 is expressed in DRG neurons of various sizes”
Response 18: We deleted and revised unnecessary or abstract words in discussion.
Q19
L193: “TRPV1, which is one of the channels involved in heat-nociception”. The same has been written in L188-189.
Response 19: Thank you for your comment. We have edited the repeated sentences.
Q20
L199: Piezo1 is blocked by non-specific inhibitors such as…
Response 20: Thank you for your comment. We have edited this sentence.
L199-201: “Activation of Piezo1 by mechanical stimuli is blocked by non-specific inhibitors such as ruthenium red, Gd3+, and the spider peptide GsTMx4 (which is a more specific inhibitor of mechanosensitive cationic channels)”
Q21
L215-224 and 245-257: I do not understand the signification of these paragraph sin the discussion. They do not deal with the subject of this article.
L258-259: The sentence “we have confirmed the physiological function of Piezo1 expression” is wrong, since no evidence on the role of Piezo1 is brought by this article.
L265: with TRPV1, which might regulate Piezo1 functions
Response 21: Thanks for your comment. We deleted and revised unnecessary or abstract words in the discussion section
Q22
L266-267: “Piezo1 channel should be considered as one of the molecular targets in mechanical hypersensitivity and nociception”. How can it be stated? The authors should be aware of the fact that concerning nociceptors, especially unmyelinated DRG neurons:
- Most small-diameter C fibers include nociceptors which are both heat and mechanically sensitive (CMHs).
- Other C fibres include heat-responsive, but mechanically insensitive “silent” nociceptors. They develop a mechanical sensitivity only in the case of injury…
One of them might correspond to the TRPV1- and Piezo1-expressing cells.
Even reworked, the discussion still needs to be ameliorated to discuss the data generated as part of this study, which is essentially a pharmacological study, not a physiological study. Indeed, nothing is tackled, experimentally speaking, on the function of Piezo1.
Response22: Thank you for your comment and we apologize for the insufficient explanation. We have deleted or revised sentences or paragraphs as a whole in the discussion section
Minor
Q23
1) L43: “of the Piezo channel family”
2) Figure 1F: 0Ca2+ + Yoda1
3) L79: in mouse DRG neurons (Figure 1A and B)
4) L123: 2.3. Pharmacological Inhibition of Piezo1 in Mouse DRG Neurons
5) L141: The Yoda1 (10 μM, 10 s)-induced intracellular Ca2+ increase
6) L162: Yoda1-induced Ca2+ responses
7) L204: Thus, Yoda1 selectively activates Piezo1 but not Piezo2
Response 23: Thank you for your comments. We revised each minor comment mentioned.
1) L42: “Piezo channel family”
2) L89: Figure 1F
3) L79: “in mouse DRG neurons (Figure 1A and B)”
4) L126: “Pharmacological Inhibition of Piezo1 in Mouse DRG Neurons”
5) L145: “The Yoda1 (10 μM, 10 s)-induced intracellular Ca2+ increase.”
6) We have deleted the sentence included this phrase.
7) We have revised discussion included this sentence.
L204-207: “A total of 21 amino acids within a minimal Yoda1 binding motif (mouse Piezo1 residues 1961-2063), also known as the agonist transduction motif (ATM), are not conserved between Piezo1 and Piezo2. Therefore, theses amino acids may form Piezo1-specific Yoda1 binding site or may influence Yoda1 binding to the Piezo1 pore.”

Round 2
Reviewer 1 Report
I have no further comments
Reviewer 2 Report
I think that the authors have answered all the questions addressed to them in a satisfactory manner. Generally speaking, there are still a number of grammatical errors, but the manuscript is better than what is was.
This manuscript is a resubmission of an earlier submission. The following is a list of the peer review reports and author responses from that submission.
Round 1
Reviewer 1 Report
In this manuscript the Authors reported the functional expression of Piezo1 channels in mouse and human DRG neurons and shown that the channel activity is inhibited by the activation of TRPV1. The paper deals with an important topic in the field of mechanical sensory transduction and nociception. Nevertheless, there are several issues which deserve further consideration.
How cell diameters were measured? Could you specify the temperature values during calcium imaging and patch-clamp experiments? Could you describe in details how Yoda1 and other substances were applied in electrophysiological and calcium imaging experiments? What is the percentage of total DRG neurons that were responsive to Yoda1 in calcium imaging experiments? How do you explain the differences in the percentage of responsive small-sized neurons in patch clamp (24.0%) and calcium imaging experiments (92.6%)? Why did you apply Yoda1 about 300 s after capsaicin in calcium imaging experiments and only after about 100 s in whole-cell experiments? How do you explain the different effect of TRPV1 activation on Piezo1 channels in electrophysiological and calcium imaging studies? For a t-test to be valid, the assumption of data distribution normality should be meet. For this reason, you should perform a preliminary check (Shapiro-Wilk test for each sample). If the hypothesis of normality has to be rejected you consider transforming your data (log-transformation) or using a non-parametric alternative. The discussion is not always pertinent and can be more concise
Reviewer 2 Report
Major Comments-
Can the authors clarify how many animals and exactly what age were used across all experiments? How many neurons were tested from each animal? Which spinal segment were the DRG collected from? Do the authors believe that the percentage of yoda1 responding neurons would change depending on the spinal segment?
The authors state they detected piezo 1 in two different human DRG, does this mean two different humans or two DRG? Additionally, what was the age and sex of the humans? How many hours/days were the neurons in culture prior to the experiment? How many neurons were screened in total, i.e. what percentage of neurons responded Yoda1?
It is unclear how the data were normalized or how intracellular calcium concentration was determined. Did the authors run a calibration experiment? The units for calcium imaging ([Ca2+]i) seems inappropriate, calcium concentration should be in molarity. It might serve better to present the data as delta F.
The authors should state exactly what statistical test they used for each graph as well as the t- or F- values, n, etc. The statistics or the way they are portrayed for the first and second and/if there was a third application of yoda1 are incorrect. The authors should use a paired-t-test or an ANOVA with RM as they are comparing within cell.
Exactly how long did the authors wait in between the sequential applications of Yoda1? Does the current ever recover?
The concentration and application time of Yoda1 seems arbitrary. Did the authors try other concentrations? Especially in the humans DRG neurons? Yoda1 may not be selective for piezo1 in human. It would be an important control to repeat the antagonist work done in Figure 3 for human.
For figure 2C, the percentage in the text does not match the graph. For example, the authors state that piezo1 is expressed in 41.7% (5/15) medium sized neurons and 15% of large-sized neurons, yet the graph indicates that there are ~30-35% medium, and ~25% large.
Do the authors have a theory as to the product difference between the mouse and human for piezo1 and gapdh in Figure 2?
The evoked calcium transient to Yoda1 in figure 3E is very different compared to other cells in previous figures. Are the authors sure the shrna KD control had no effect on the cells? Perhaps did KCl at the end?
The authors already show that sequential application of yoda1 decreases the evoked current (Figure 1 A &B), how can the authors be sure that the result they see from 3 neurons in figures 4C & D are due to trpv1 activation and not just from sequential hits of yoda1?
Do the authors have a theory as to the complete lack of current evoked by yoda1 after capsaicin, yet still induces a clear calcium transient?
Are the time-stamps for drug application in figure 4A and C accurate? If so, there is a significant delay in the evoked current. What was the rate of flow? The current trace evoked by yoda1 in figure 1 is much larger than the current evoked in figure 4A and C.
In the discussion, the authors state that piezo1 “…primarily co-expressed with TRPV1 in small- and medium-sized DRG neurons.” Is this referring to data collected in Figure 4? Figure 4 does not indicate cell-size whatsoever. According to figure 2C, piezo1 is expressed in ~25% of large diameter neurons.
Authors should reread their discussion; a number of the paragraphs are irrelevant to the current study. For example, the authors did not study piezo2 in any capacity, yet there is a whole paragraph discussing piezo2.
Minor comments-
Figure 1 legend, the authors use 50mM KCl but Figure 4 legend, authors state they used 50uM KCl? Additionally, how long was the KCl applied for?
Figure 3 graphs should be larger.
When referring to figure 3 values in results section 2.3, it is unnecessary to go out to more than 1 decimal place.
The evoked calcium transient to Yoda1 in figure 3E is very different compared to other cells. Are the authors sure the shrna KD control had no effect on the cells? Perhaps a KCl at the end?
In Methods 4.6 calcium imaging, authors state they loaded with fura-2 then plated the neurons onto a poly-D lysine coverslip? As this reviewer understands it, this mean the neurons were stripped after they loaded with fura then replated?
In Methods 4.9 Western blot, authors state they used GAPDH, but show Bactin in Figure 2B.
The first sentence in the discussion, the authors write, “To date, no previous study has reported the expression or function…” The expression has been reported as the authors state in their introduction by the Hamil group, at least the mRNA expression. Authors should clarify, functional expression.